# Adverse Food Reactions in Inflammatory Bowel Disease: State of the Art and Future Perspectives

**DOI:** 10.3390/nu16030351

**Published:** 2024-01-25

**Authors:** Ivan Capobianco, Federica Di Vincenzo, Pierluigi Puca, Guia Becherucci, Maria Chiara Mentella, Valentina Petito, Franco Scaldaferri

**Affiliations:** 1Dipartimento di Medicina e Chirurgia Traslazionale, Università Cattolica del Sacro Cuore, 00168 Rome, Italy; federica.divincenzo30@gmail.com (F.D.V.); pgpuca@gmail.com (P.P.); franco.scaldaferri@policlinicogemelli.it (F.S.); 2UOC Nutrizione Clinica, Dipartimento di Scienze Mediche e Chirurgiche Addominali ed Endocrino Metaboliche, Fondazione Policlinico Universitario Agostino Gemelli IRCCS, 00168 Rome, Italy; guia.becherucci01@icatt.it (G.B.); mariachiara.mentella@policlinicogemelli.it (M.C.M.); 3IBD Unit, UOC CEMAD Centro Malattie dell’Apparato Digerente, Dipartimento di Scienze Mediche e Chirurgiche Addominali ed Endocrino Metaboliche, Fondazione Policlinico Universitario Agostino Gemelli IRCCS, 00168 Rome, Italy; valentina.petito@policlinicogemelli.it

**Keywords:** food allergies, food intolerances, inflammatory bowel disease

## Abstract

Limited knowledge is available about the relationship between food allergies or intolerances and inflammatory bowel disease (IBD). Clinicians frequently encounter patients who report food allergies or intolerances, and gastroenterologists struggle distinguishing between patients with organic disorders and those with functional disorders, which the patients themselves may associate with specific dietary components. This task becomes even more arduous when managing patients with significant underlying organic conditions, like IBD. The aim of this review is to summarize and emphasize any actual associations between food allergies and intolerances and inflammatory diseases, such as ulcerative colitis and Crohn’s disease. Through a narrative disceptation of the current literature, we highlight the increased prevalence of various food intolerances, including lactose, fructose, histamine, nickel, and non-celiac gluten sensitivity, in individuals with IBD. Additionally, we explore the association between increased epithelial barrier permeability in IBD and the development of food sensitization. By doing so, we aim to enhance clinicians’ awareness of the nutritional management of patients with IBD when facing complaints or evidence of food allergies or intolerances.

## 1. Introduction

Inflammatory Bowel Disease (IBD) is a group of organic conditions that affects ap-proximately 1% of the population, encompassing two main forms: ulcerative colitis (UC) and Crohn’s disease (CD). While the clinical presentation may vary between UC and CD, two primary symptoms are consistently observed: abdominal pain and diarrhea (with or without blood) [1].

Symptoms such as diarrhea, flatulence, abdominal distension, and abdominal pain are frequently reported by many patients receiving gastroenterological assessment. Clinicians often encounter patients who associate these symptoms with specific foods. Establishing a clear connection between the symptoms experienced and the food consumed can be quite challenging. Dietary restrictions are also common among patients diagnosed with IBD, based on the belief that certain foods are intolerable and may exacerbate underlying disease symptoms. A Danish study involving 132 patients with IBD revealed that 66% of CD patients and 64% of UC patients often restricted their food intake due to the reported experiences of food intolerance. This was observed irrespective of factors such as previous surgeries, disease activity, or location [2].

Many other studies have shown that dietary beliefs exert a significant impact on the social lives of individuals with IBD. Most IBD patients believe that certain foods could exacerbate their symptoms and trigger disease relapses. This leads to dietary restrictions, especially during disease flares and symptoms exacerbation.

The most commonly eliminated foods include vegetables, fruits, tomatoes, legumes, spicy foods, milk and dairy products, fried foods, alcohol, coffee, and high-fat items. These aliments are reported to worsen symptoms [3,4].

Conversely, yogurt, rice, and bananas were frequently reported to have an ameliorative effect on symptoms [4].

Gaining a deeper understanding of the impact of food intolerances on patients with IBD is crucially important. Recent research has shown that levels of fecal calprotectin, a widely used marker for distinguishing between functional and organic bowel disorders, as well as assessing the severity of IBD, may be elevated in patients with lactose intolerance, fructose malabsorption, or histamine intolerance [5].

Considering the significance of nutritional status in individuals with IBD and the potential adverse effects of self-imposed dietary restrictions, interrogating the current scientific literature to determine if a genuine correlation exists between IBD and food intolerances is of paramount importance. This examination aims to ascertain whether the symptoms experienced by patients are truly influenced by the foods they consume or if they are independent of dietary factors and possibly correlated to mechanisms associated with their underlying condition. 

These findings underscore the importance of avoiding misinterpretations, which can result in misguided therapeutic strategies and profoundly impact the nutritional well-being of patients. Additionally, they highlight the imperative for further research to explore innovative interventions and personalized dietary strategies, with the aim of enhancing the quality of life for individuals with IBD.

## 2. Methodology

Databases, including PubMed, Scopus, and Web of Science, were systematically queried using key terms such as “food allergies”, “food intolerances”, and “Inflammatory Bowel Disease”. The search focused on articles published between 2010 and 2023. Inclusion criteria consisted of studies investigating the prevalence, mechanisms, and clinical implications of lactose intolerance, food allergies, and various food intolerances in individuals with IBD. Our review mainly encompasses clinical trials, observational studies, systematic reviews, and meta-analyses. Conversely, the selection process did not involve screening titles, abstracts, and other kinds of unpublished material. Recent and high-impact publications were emphasized to ensure the inclusion of the latest evidence. The synthesis of findings from diverse studies informs the insights presented in this narrative review.

## 3. Food Allergies and Intolerances

An adverse reaction to food is defined as an undesirable response that occurs following the consumption of a particular food. These adverse reactions are classified based on the pathogenic mechanisms involved in various reactive syndromes. They can be categorized as follows (Figure 1):

Reactions of toxic nature: these are caused by substances that can contaminate foods. They are dose-dependent and affect all individuals exposed to sufficiently high doses.

Hypersensitivity reactions: these reactions are generally unpredictable and affect individuals who are predisposed. Hypersensitivity reactions can be further divided into two types:Reactions mediated by the immune system (food allergies).Reactions not mediated by the immune system (food intolerances) [6,7].

Food allergies are primarily mediated by Immunoglobulin E (IgE). The prevalence of food allergies is estimated to be approximately 8% in children and 10% in adults [8]. While any food has the potential to trigger an allergic reaction, commonly implicated foods include milk, eggs, peanuts, fish, shellfish, and various plant-based foods [9]. Cross-reactivity syndromes can also arise, wherein the clinical manifestation involves the simultaneous occurrence of two or more allergies due to the structural similarity of allergenic proteins [10]. For instance, individuals with cow’s milk allergy, currently the most prevalent allergy [11], may also experience allergic reactions to goat and sheep’s milk, as well as to beef, due to the presence of serum albumins and immunoglobulins. Allergic reactions to crustaceans and mollusks are often co-present due to cross-reactivity, as it happens for various finned bony fishes. Grains typically have low cross-reactivity rates, while other foods tend to vary in this regard [12]. These allergies can have significant clinical and social consequences, adversely affecting patients’ quality of life and, in severe cases, even leading to fatalities. 

Food intolerances can be further classified into three categories:

Enzyme deficiencies: these intolerances arise from the inability to metabolize certain substances present in food. They can result from rare congenital deficiencies (such as phenylketonuria and galactosemia); common deficiencies (like lactase deficiency and G6PDH deficiency); or gastrointestinal tract diseases, such as infections, celiac disease, or Crohn’s disease. Sugars, including lactose, fructose, trehalose, sucrose, and sorbitol, are primarily responsible for this type of intolerance. When these sugars are not properly absorbed, they reach the colon and undergo fermentation by bacteria. This fermentation process leads to the production of gas (hydrogen, carbon dioxide, methane) and short-chain fatty acids, with a consequent osmotic effect that attracts liquid into the intestinal lumen. These mechanisms can result in symptoms like bloating, distension, abdominal pain, and diarrhea [13].

Pharmacological intolerances: these intolerances can be direct or indirect. Direct pharmacological intolerances occur when a naturally occurring chemical substance in food, such as vasoactive amines, triggers a reaction. Indirect pharmacological intolerances happen when certain foods induce the release of mediators (i.e., histamine) from cells. There are also undefined pharmacological intolerances whose exact mechanism is unknown, often associated with additives. Foods rich in histamine (such as fermented cheeses, salami, oily fish, tuna, spinach, and tomatoes), caffeine (found in coffee, chocolate, tea, and cola), phenylethylamine (found in chocolate, cheese, and red wine), and tyramine (found in potatoes, tomatoes, cabbage, and tuna) can cause pharmacological intolerances [13].

Undefined intolerances: these intolerances are categorized as undefined when the precise mechanism is unknown. Reactions to food additives (such as preservatives, dyes, sweeteners, and antioxidants) fall into this category. Foods like canned meats, vegetables, canned meat products, syrups, fruit juices, and others may contain these additives and can trigger reactions [7,13,14].

Table 1 summarizes the differences between food allergies and intolerances.

## 4. Food Intolerances and IBD

### 4.1. Lactose Intolerance and IBD

Lactose is a disaccharide composed of galactose and glucose, playing a vital role in providing calories through milk for most mammals. It serves as a primary source of calories in infancy and remains an essential component of the diet for populations that during adulthood retain the ability to digest this carbohydrate (Lactose Persistent—LP) [15].

Lactose intolerance (LI) is identified by the onset of abdominal symptoms like abdominal pain, bloating, and diarrhea following the consumption of lactose by an individual with lactose malabsorption (LM) [16].

LM refers to the inability to absorb lactose in the small intestine, which is commonly a result of lactase downregulation after infancy. In Caucasians, lactase non-persistence (LNP), associated with the LCT−13′910:C/C genotype, is the prevailing mechanism behind this condition [16]. Additionally, conditions that impact the mucosal integrity of the small bowel, including infections and other factors, can potentially result in secondary lactase deficiency and subsequent lactose malabsorption [17].

The relationship between LI and IBD, including UC and CD, is a controversial topic that is currently under investigation and research.

The existing evidence has yielded mixed results, and a definitive answer has not been established so far.

In a comprehensive meta-analysis conducted by Szilagyi et al., which included 17 studies and 1935 IBD patients, it was observed that there is an overall higher prevalence of LI in both UC and CD, irrespective of the method used for diagnosis. However, upon subgroup analysis, the increased risk was statistically significant only for patients with CD and not for those with UC. Among CD patients, those with small bowel involvement showed a significant association with LI, whereas those with only colon involvement did not exhibit the same trend [18]. These findings support the idea that ileal disease may contribute to secondary lactase deficiency in these individuals.

Furthermore, the findings of the meta-analysis suggest that both individuals with lactase deficiency (hypolactasic) and those without lactase deficiency (normolactasic) may be equally affected by IBD. This is consistent with previous research analyzing genetic polymorphisms between IBD patients and healthy individuals, where no significant differences were observed [19].

According to multiple studies, the prevalence of LI in patients with IBD does not appear to differ significantly from the one of health controls [20,21,22]. It seems to depend on the epidemiological population distribution of lactase persistence or non-persistence.

However, in some studies, lactase persistence appears to be linked with CD. In a New Zealand Caucasian population, individuals with a homozygous T/T genotype exhibited a significantly higher risk of developing CD compared to those with a homozygous C allele (OR = 1.61, 95% CI = 1.03–2.51). Furthermore, there was a noteworthy increase in the frequency of the T allele among CD patients (OR = 1.30, 95% CI = 1.05–1.61, *p* = 0.013), indicating that the T allele associated with lactase persistence was correlated with an elevated risk of CD [23].

The rise in LNP within populations has also been associated with a reduced risk of CD. A study involving populations from 26 countries worldwide demonstrated that the prevalence of CD decreased as the prevalence of LNP increased (*p* < 0.01) [24,25].

In a recent article, the prevalence of LI (evaluated using hydrogen breath test) among IBD patients in clinical remission was comparable to the one of the control group (64.8% vs. 62.3%). Additionally, the investigation of the wild-type polymorphism associated with LNP showed no significant difference between the IBD population and the control group (85.2% vs. 87%) [21]. These findings suggest that approximately one-third of IBD patients are not lactose intolerant, which aligns with the proportion found in healthy individuals.

However, a recent American study, encompassing almost 600,000 patients with IBD extrapolated from the National Inpatient Sample (NIS) database, revealed that patients with IBD had nearly triple the risk of lactose intolerance compared to the control group. The authors emphasize the significance of screening for LI in IBD patients, given the robust association they uncovered [26].

This result is inconsistent with the studies mentioned earlier. This misalignment could be attributed to the fact that the NIS primarily relies on data from hospitalized individuals, possibly leading to a higher proportion of individuals with severe illness. Furthermore, the NIS database does not provide specifics about whether the diagnosis of LI was confirmed through specific testing. This circumstance might have influenced the generalizability of the findings. 

It is ascertained that during active stages of disease, both UC and CD patients experience an improvement of symptoms when placed on lactose-free diets [27]. In the case of UC, disruption in the small bowel mucosa could lead to reduced disaccharidase content within the epithelium, with consequent decreased levels of the lactase enzyme [28,29]. These observations align with the concept that LM in these individuals might be a consequence of their intestinal disease, as previously discussed.

Certainly, in light of what was mentioned earlier, it is clear that lactose is not the only factor to be considered when assessing the “tolerance” of dairy products of patients with IBD.

The estimated prevalence of sensitivity to dairy foods in individuals with IBD is approximately 10% to 20%, and apparently, at least 20% of patients with active ulcerative colitis and 30% of patients with Crohn’s disease could experience benefits from a dairy-free diet, irrespective of lactose intolerance [30].

In a study involving 165 adults with CD, the authors found that dairy products had no significant effect on self-reported CD symptoms for most individuals. However, high-fat foods were more frequently reported to exacerbate perceived CD symptoms. Interestingly, the lactose content of individual dairy products did not significantly influence self-reported CD symptoms for the majority of patients [31]. Additionally, colonic inflammation was more frequently associated with an increase in reported adverse effects of consuming dairy products in comparison to ileal involvement [31].

Many diets recommended for patients with IBD tend to exclude or significantly reduce the consumption of milk and dairy products. Notorious examples are the Crohn’s disease exclusion diet (CDED); the IBD-Anti-inflammatory diet (IBD-AID); the low fermentable oligosaccharides, disaccharides, monosaccharides, and polyols diet (LOW-FODMAP); and the specific carbohydrate diet (SCD) [32,33,34,35,36]. Many of these diets seem to alleviate functional symptoms in IBD patients, likely without affecting the organic aspects of disease, such as mucosal inflammation [37,38].

It is worth considering that the consumption of dairy products is not a risk factor for IBD [39]. Individuals who consume milk had significantly reduced odds of developing CD (OR: 0.30, 95% CI: 0.13–0.65), suggesting a protective effect of dairy product consumption [40]. Furthermore, dairy foods are a crucial source of protein and the primary dietary source of calcium. Their avoidance increases the risk of bone demineralization and osteoporosis, well-known complications of CD, which can be further exacerbated by the use of steroid medications (common in IBD treatment) [25,41,42].

Therefore, if lactose intolerance is confirmed, it is advisable to recommend the use of low-lactose or lactose-free dairy products, like milk and yogurt. However, if dairy foods are still not tolerated, non-dairy sources of calcium become essential to prevent the risk of further malnutrition in predisposed individuals [43].

### 4.2. Fructose Intolerance and IBD

Although lactose intolerance is widely acknowledged as a source of nonspecific gastrointestinal complaints, the lesser-known malabsorption of other carbohydrates, such as fructose, also plays a central role. 

Fructose, a hexose sugar, is commonly consumed in Western diets and occurs naturally in fruits like apples, peaches, pears, and oranges [44]. Its absorption occurs with limited capacity in the small intestine through facilitated diffusion [45]. In case fructose remains unabsorbed and reaches the colon, it undergoes fermentation by anaerobic colonic flora. This process can lead to an osmotic effect causing distention of the small intestine, resulting in symptoms like abdominal pain, bloating, discomfort, and diarrhea [46,47].

The molecular and genetic triggers for fructose malabsorption (FM) remain unidentified. The role of GLUT5, the primary intestinal fructose transporter, in the pathophysiology of FM is also not yet clearly understood [48,49]. Fructose hydrogen breath testing (BHT) is employed in clinical settings to diagnose malabsorption. However, a comprehensive validation of this test with established testing conditions and diagnostic thresholds is still lacking [50,51,52].

Numerous studies suggest a higher occurrence of fructose intolerance (FI) in individuals with functional gastrointestinal disorders [53,54,55]. Yet, the connection between FI and organic conditions such as IBD remains uncertain.

Barrett et al. conducted a comparison of FM prevalence between healthy individuals and those with chronic intestinal disorders. Within this analysis, CD displayed a higher occurrence of FM (61%) compared to other groups (33–44%, *p* < 0.05) [56].

In animal models, particularly rabbits, it was observed that the inflammatory mediators, particularly TNF alpha, downregulate the GLUT5 receptor [57,58]. This led to postulating that inflammation might be responsible for GLUT5 receptor downregulation and consequently cause FM in IBD patients. 

However, recent research indicates that the GLUT5 receptor is not downregulated in cases of FM. No significant differences were found in the results of the H2 breath test between patients undergoing anti-TNF therapy and patients receiving other medications. Furthermore, the clinical response to a fructose-restricted diet in fructose-intolerant patients was not associated with differential GLUT5 receptor expression [59].

In a recent study, patients with active and inactive IBD were compared to healthy controls in terms of FM prevalence; no variations in FM were detected among groups (35/44 (79.6%) active IBD, 59/80 (73.8%) inactive IBD, and 66/81 (81.5%) healthy controls). Nonetheless, abdominal pain was more frequent in patients with active IBD compared to those with IBD in remission, and diarrhea was more common in patients with active IBD than in healthy controls [59].

This supports the hypothesis that increased intestinal sensitivity in individuals with IBD might contribute to heightened symptoms during active phases, regardless of the prevalence of FM.

### 4.3. Histamine Intolerance and IBD

First discovered more than a century ago, histamine (2-[3H-imidazol-4-yl] ethanamine) derives from the decarboxylation of the amino acid L-histidine through the L-histidine decarboxylase (HDC) [60,61,62]. Stored within the secretory granules of basophils, mast cells, and histaminergic neurons by the Golgi apparatus, histamine is detected in almost all human tissues [63]. Moreover, macrophages, neutrophils, eosinophils, B and T lymphocytes, Langerhans cells, dendritic cells, as well as intestinal epithelial cells (IECs), can serve as additional sources of histamine release [64,65,66].

Histamine-releasing cell degranulation usually occurs when a specific antigen binds to the FcepsilonRI (FcεRI) receptor on the surface of basophils and mast cells, or in reaction to nonimmune stimuli, such as tissue injury [67]. Diamine oxidase (DAO) is the principal enzyme responsible for histamine catabolism, converting histamine to imidazole acetaldehyde [68]. Another enzyme involved in the catabolism of histamine is Histamine-N-methyltransferase (HNMT), which transfers a methyl group from S-adenosyl-L-methionine (SAM) to histamine to produce N-methylhistamine (NMH), which is further converted to N-methylimidazole acetic acid [69].

The gut microbiota includes many Gram-positive and Gram-negative bacteria, such as *Escherichia coli*, *Lactobacillus reuteri*, *Lactobacillus vaginalis*, *Proteus vulgaris*, *Proteus milabilis*, *Hafnia alvei*, *Morganella morganii*, *Enterobacter aerogenes*, *Raoultella ornithinolytica*, *Raoultella planticola*, *Pseudomonas fluorescens*, *Citrobacter freundii*, and *Photobacterium damselae*, that contribute to the metabolism of histamine through the production of HDC and consequent release of the biogenic amine histamine [70]. An acidic environment increases the bacterial synthesis of amino acid decarboxylase, which raises the local pH around the bacteria and protects them against a chloride-rich acidic environment. Moreover, the presence of fermentable carbohydrates and oxygen, the redox potential of the medium, temperature, and the concentration of sodium chloride (NaCl) all affect the expression and activity of decarboxylases in bacteria [71].

Besides the endogenous source, histamine can also originate from an array of exogenous sources, including gut microbiota, and several foods, especially those that have been fermented, aged, microbiologically altered, or otherwise processed, including seafood, wine, cheese, soy sauce, sauerkraut, alcoholic beverages, and jerky, where the main route for histamine formation is histidine decarboxylation mediated by the bacterial HDC [70,71,72,73,74]. In addition to histamine, these types of food may also contain cadaverine (pentane-1,5-diamine) and putrescine (1,4-diaminobutane). These substances compete with histamine for DAO binding sites, leading to an impact on histamine metabolism. As a result, histamine accumulates, causing the development of histamine intolerance (HIT). HIT is a non-immune reaction characterized by the accumulation of histamine due to a reduced capacity for histamine degradation [73,75]. Clinical manifestations of HIT consist of a wide range of nonspecific gastrointestinal symptoms, such as diarrhea, abdominal pain, bloating, flatulence, postprandial fullness, constipation, nausea, and emesis, as well as extra-intestinal symptoms, due to the ubiquitous distribution of the four histamine receptors (HRs), including rhinitis, sneezing, headache, dizziness, hypotonia, tachycardia, flush, pruritus, eczema, urticaria, and swelling [76].

The pleiotropic effect of histamine is mediated by its binding to one of the four histamine receptors (HRs). All HRs, except H3R, were found to be expressed in the human intestine [77], with demonstrated pro-inflammatory outcomes for histamine receptor 1 and 4 (H1R and H4R) [78,79] and anti-inflammatory properties for histamine receptor 2 (H2R) [80,81] in IBD patients. Indeed, histamine can suppress the pro-inflammatory response caused by the bacterial activation of toll-like receptor (TLR)2, TLR4, TLR5, and TLR9 via H2R activation [82], while H1R and H4R signaling synergizes cAMP buildup and MAPK activation to enhance the expression of pro-inflammatory genes [79].

Histamine and HRs have been identified as two key elements in the signaling pathways influencing the development of IBD [82,83]. Several studies showed an increased number of mast cells in the submucosa of ileal samples from patients with CD [84,85], as well as in patients in the active stage of UC [86]. Furthermore, mast cells were on average more abundant in inflamed colonic tissue compared to unaffected and noninflamed mucosa in patients with IBD (2000 per milligram (mg) of tissue vs. 1500 per mg of tissue in UC and 1700 per mg of tissue vs. 1250 per mg of tissue in CD) [87]. Previous studies showed that mast cells collected from surgical colonic specimens of patients with active CD and UC release more histamine compared to normal colonic mast cells [88,89,90]. Furthermore, mast cells have been found to play a role in the pathogenesis of dextran sulfate sodium (DSS)-induced colitis in mice models [91,92]. Notably, Winterkamp et al. revealed that patients with active UC and CD presented an increased urinary excretion of the stable histamine metabolite NMH, compared to inactive IBD or non-IBD controls, and the urinary NMH excretion strongly correlated with the endoscopic CD activity assessed by the CD Endoscopic Index of Severity (r2 = 0.70, *p* < 0.0001) [93].

Interestingly, in patients affected by IBD, four single-nucleotide polymorphisms (SNPs) of DAO have been detected, C47T (Thr16Met), C995T (Ser332Phe), C4106G (His646Asp), and G-4586T, as well as three HNMT SNPs, A595G, C314T, and A939G, resulting in amino acid substitution, which decreased protein activity and stability, thereby leading to impaired histamine metabolism regulation in the inflamed intestine [94,95,96]. Moreover, another investigation on a larger cohort of UC patients detected Thr105Ile and His645Asp SNP in HNMT and His645Asp in DAO, also suggesting a correlation between His645Asp SNP in DAO and the severity of UC [97]. Curiously, the DAO activity in blood was found to be significantly lower in patients with CD and UC compared to the control population, thus suggesting its potential importance as a marker of intestinal permeability and supporting the hypothesis that individuals with SNP in DAO may be more susceptible to the development of IBD [98].

To date, there are no data available regarding the incidence and prevalence of HIT in an IBD population, nor have any studies analyzed the proportion of gastroenterological symptoms attributable to histamine intolerance in an IBD patient population. Indeed, as we have previously detailed, SNPs associated with the impaired function of enzymes involved in histamine metabolism have been identified in IBD patients, and the screening for such and other DAO SNPs, associated with impaired DAO activity (i.e., rs1049742, rs10156191, and rs1049793), is part of the diagnostic algorithm of HIT [76]. Therefore, it is assumable that the IBD population might be at higher risk of HIT development as a result of consuming foods high in histamine; in the same way, in some IBD patients, nonspecific gastroenterological symptoms not ascribable to the inflammatory bowel disease could result from an underlying HIT condition. However, further studies are needed to corroborate this hypothesis in order for physicians to advise IBD patients with such nonspecific HIT-related symptoms on the consumption of a low-histamine diet or DAO supplementation [76].

### 4.4. Non-Celiac Gluten Sensitivity and IBD

Non-celiac gluten sensitivity (NCGS) is a condition characterized by both intestinal and extra-intestinal symptoms related to the ingestion of gluten-containing foods in the absence of a diagnosis of celiac disease or wheat allergy [99,100]. Since there are currently no specific diagnostic tests or biomarkers for the diagnosis of NCGS, this condition may fall under the broad spectrum of irritable bowel syndrome (IBS) [101]. Usually, the diagnosis of NCGS is self-diagnosed by patients who report relief of symptoms such as bloating, discomfort or abdominal pain, changes in bowel habits, fatigue, headache, or even depression after starting and adhering to a gluten-free diet (GFD). In 2011, a double-blind, randomized, placebo-controlled rechallenge trial was conducted to investigate the hypothesis that gluten may induce gastrointestinal and non-gastrointestinal symptoms in patients with no diagnosis of celiac disease. Enrolled patients who met a GFD due to NCGS were given either a gluten or placebo in the form of two slices of bread and one muffin daily, in addition to a GFD over 6 weeks. The study showed that patients in the gluten arm were more likely to report symptoms, including bloating, abdominal pain, dissatisfaction with stool consistency, headaches, and fatigue, thereby corroborating the existence of NCGS [102].

To date, incidence and prevalence rates of NCGS in patients with IBD have not been extensively studied. In a large cohort involving 102 IBD patients from a tertiary-care center, NCGS was described in 23.6% of CD and 27.3% of UC. The patients with NCGS were considerably younger (34.9 vs. 42.4 years, *p* = 0.04) than their counterparts without gluten sensitivity. Patients with NCGS were found to follow a GFD at a considerably higher rate (64% vs. 5.2%, *p* < 0.001); nonetheless, there was no significant difference in the reported gastrointestinal symptoms between IBD patients with and without NCGS [99]. Conversely, Aziz et al. discovered that IBD patients with NCGS most frequently complained of headaches, lethargy, bloating, diarrhea, and abdominal pain or discomfort following gluten consumption [101].

Assessing IBD-related factors, gluten sensitivity was found to be significantly associated only with having had an IBD flare-up within the previous 60 days (adjusted odds ratio (aOR) 7.4; 95% CI 1.6–34.1), fibrostenotic disease in CD (aOR 4.7; 95% CI 1.1–20.2), and dermatological extraintestinal manifestations (aOR 5.5; 95% CI 1.2–24.1). With respect to the former, the authors suggested that gluten sensitivity might be transient during the improvement of the intestinal barrier after IBD flare-up [99].

The cross-sectional study by Aziz et al. produced similar findings, demonstrating a prevalence of NCGS in IBD patients of 27.6%, with a significantly higher rate of NCGS in CD patients with fibrostenotic disease (40.9% vs. 18.9%, *p* = 0.046) and a higher mean Crohn’s Disease Activity Index (CDAI) score (228.1 vs. 133.3, *p* = 0.002) [101]. It is likely that, in patients with fibrostenotic CD, NCGS results from the presence of fermentable carbohydrates in gluten-containing foods. These aliments causing an increase in the luminal concentration of gas following their intestinal fermentation could ultimately result in increased abdominal pain and discomfort in patients with intestinal stenosis [99].

To date, there is a lack of studies evaluating the severity of intestinal inflammation and the frequency of flare-ups in patients with NCGS and IBD [103]. Notably, in TNFΔARE/WT mice, a gluten-fortified diet resulted in the development of chronic ileitis; increased expression of pro-inflammatory cytokines, including tumor necrosis factor (TNF), interferon-gamma, and interleukin (IL)-15; as well as decreased expression of occludin, a tight junction protein involved in maintaining intestinal permeability [104].

Given the role of gluten in intestinal inflammation, several studies investigated and discussed the potential role of GFD in alleviating gastrointestinal symptoms in IBD patients. This is particularly important considering that many IBD patients have reported that their symptoms are linked to specific foods, especially gluten [105]. Nonetheless, prospective controlled studies on GFD in IBD patients are lacking, and only survey-based cross-sectional studies and study questionnaires exhibited that a GFD may improve gastrointestinal symptoms and/or influence disease severity in patients with IBD [3,99,101,105,106].

Interestingly, Morton et al. reported that consuming gluten-containing bread was associated with the occurrence of symptoms’ exacerbations in a cohort of 233 IBD patients who answered a self-administrated questionnaire, whereas gluten-free bread was well tolerated [107].

Accordingly, Herfarth and colleagues performed a cross-sectional study where a dietary questionnaire on adherence to GFD was administered to 1647 patients affected by IBD. According to the authors, 81 patients (4.9% of the total enrolled patients) had a previous diagnosis of NCGS, while 314 participants (19.1%) reported having tried a GFD in the past, and 135 participants (8.2%) reported currently following a GFD [105]. In conclusion, the authors suggest a trial of GFD in IBD patients with considerable gastrointestinal symptoms as a safe and sometimes effective treatment option, although further longitudinal studies are required to explore the mechanism of gluten sensitivity in IBD [105]. Overall, 38.3% of patients who attempted a GFD reported an improvement in the severity and frequency of disease flares, 23.6% of them required fewer IBD medications after the GFD trial, and 65.6% of all patients showed amelioration of at least one of their gastrointestinal symptoms (e.g., bloating, abdominal discomfort, diarrhea, abdominal pain, nausea), as well as fatigue [105].

Conversely, Zallot et al., in a cohort of 244 IBD patients, demonstrated with a 14-item questionnaire that 9.5% of the enrolled patients considered GFD to be beneficial for symptom relief during an IBD flare-up, but only 1.6% actually made the decision to follow a GFD during a flare-up of the disease [3]. Similarly, Schreiner et al., in a study on 1254 individuals affected by IBD, reported no differences in disease activity, complications, surgery, and hospitalization rates between patients following a GFD and those on a free regimen. Furthermore, the authors emphasized that patients on a GFD had lower psychological well-being [106].

In conclusion, due to heterogeneity among the studies and disparate study objectives, it is not feasible to draw trustworthy recommendations on the impact of a GFD on the course of the disease or for the control of IBD symptoms in patients with or without gluten sensitivity. Consequently, further studies are warranted to assess the feasibility of advocating a GFD in clinical practice in IBD patients who complain of IBS-like symptoms.

## 5. Food Allergies and IBD

In the realm of gastrointestinal disorders, the coexistence of food allergies and IBD has long perplexed medical professionals and patients. As individuals with IBD navigate the intricate landscape of dietary choices, the added complexity of potential food allergies presents a significant challenge. While the complex interplay between these two conditions continues to unfold, understanding the mechanisms of their relationship is essential for effective management and improved quality of life, as well as for a better comprehension of pathogenetic mechanisms underlying IBD.

Common pathogenetic mechanisms of IBD and allergic manifestations could represent the substrate of such association. The chronic inflammation of IBD brings damage to the gut barrier, leading to increased exposure of mucosal immune cells to potentially allergenic elements (mainly proteins). Furthermore, Th2-polarized inflammation together with reduced Treg activity could favor the onset of allergic manifestations in patients with IBD [108]. The role of epithelial barrier disfunction is underlined by the evidence that loss-of-function alleles of filaggrin, a protein involved in the epithelial cell-to-cell adhesion, lead to increased susceptibility to eczema and asthma in IBD [109]. Some authors even postulate that food allergies could initiate colitis. In fact, healthy mice (BALB/c) repeatedly exposed to ovalbumin (a common food allergen) conjugated with cholera toxin develop heavy infiltration of inflammatory cells (dendritic cells in particular) in the colon mucosa; weight loss; increases in myeloperoxidase, tumor necrosis factor-α, interleukin-4, and mucosal OVA-specific IgE (proving sensitization to ovalbumin); as well as severe compromising of the epithelial barrier [110]. The consequent inflammation in the colon was mainly Th2-based, with a significant increment of IL4 among cytokines. It has to be underlined that specific adjuvants (cholera toxins in this case) are necessary to trigger the sensitization and the inflammatory process [110].

Whether the presence of an IBD could lead to increased sensitivity to foods is uncertain and could vary based on factors such as ethnicity or genetic background. In a cohort of Norwegian patients, UC but not CD was associated with higher levels of sensitization to foods, assessed in terms of specific circulating IgE [111]. On the contrary, a preliminary report on an Iranian pediatric cohort detected no significant differences between patients with UC and patients with CD in terms of cow milk allergy, atopic dermatitis, or positivity on multiple (>1) skin prick tests [112]. In a very recent Polish study, a pediatric cohort of newly diagnosed IBD was investigated. Both in CD and UC, the presence of elevated circulating IgE levels was associated with more complicated disease. In particular, in CD, elevated total IgE was associated with weight loss, rectal bleeding, and ASCA IgG positivity; in UC, this association was detected with abdominal pain and weight loss [113].

A role in the development of alimentary allergies could be played by circulating IgG, as well. This has been first demonstrated in murine models and then partially confirmed in humans. In facts, colitis murine models, such as IL10 knock-out mice, have shown higher levels of IgG against foods compared to controls; furthermore, the administration of beta-conglycinin, identified as an antigenic food protein, induced CD4(+) T-cell production of interferon-γ and IL-17 in the same murine model, leading to worse and more severe colitis [114]. Xiao et al. investigated circulating IgGs in patients with IBD vs healthy controls, detecting higher titers of antibodies against egg, milk, wheat, corn, rice, tomato, codfish, and soybean antigens in patients with CD compared with UC patients and healthy controls (*p* < 0.01). The levels of total serum IgG and IgE were also significantly higher in CD patients than in healthy controls (*p* < 0.01) [115]. Similar retrospective studies confirm this result, detecting a high prevalence of serum IgG antibodies to specific food allergens in patients with IBD [116]. A specific IgG subtype could be involved. A Turkish study showed higher levels of IgG4 to salmon, millet, and onion in patients with CD, while high titers of IgG4 against cuttlefish and onion were detected in UC patients [117].

However, in contrast to the earlier notion, other studies have presented evidence that challenges the association between circulating IgG levels and the prevalence of food allergies in individuals with IBD [118].

As emerges from the above-mentioned studies, the difference in sensitivity to foods between UC and CD is a subject of interest and increasing scrutiny but still without certain origin. In our judgement, two main factors explain this difference: gut microbiota and immune homeostasis. Beyond environmental factors, UC and CD exhibit significantly divergent alterations, both in gut microbiota features and immune signatures. If the allergic reaction has to be driven to the immune system ultimately, the interaction with environmental factors and gut microbiota must be taken into account [119,120].

In conclusion, the growing evidence linking food allergens to IBD highlights the importance of personalized dietary management. Further research is necessary to unravel the complexities of this relationship, with the potential to pave the way for more targeted therapeutic approaches, ultimately improving the quality of life for those affected by these interconnected conditions.

### Nickel and IBD

Nickel is a transition metal with the atomic number 28. It is the fifth most common element on Earth and is primarily found in nature in the form of oxides, sulfides, and silicates [121].

Nickel is ubiquitous in our environment, and human exposure to nickel is inevitable. Topical skin exposure to nickel occurs through metallic objects, household products, cosmetics, jewelry, metal on clothing (buttons, zips), toys, coins, utensils (keys, needles, electronic cigarettes), electronic devices, dental materials, and more. Meanwhile, systemic exposure can occur through food, surgical implants, and dental materials [121,122,123,124]. Speaking of food, nickel naturally occurs in drinking water and various food items, making it challenging to be avoided in our diet. Examples of nickel-rich foods include chocolate, legumes, shellfish, grains, nuts, and canned food [125].

Nickel is the most common cause of contact allergy in the general population and is the most frequently detected allergen in patients undergoing patch testing for suspected allergic contact dermatitis (ACD) [126].

Nickel allergy is characterized by a cell-mediated delayed-type hypersensitivity reaction (Type IV) to nickel ions. This response involves the activation of both the innate and adaptive immune systems in a complex manner [127]. Dendritic cells (DCs) play a significant role in this process, leading to the stimulation of the IKK2-nuclear factor-κB cascade, among other pathways. The degree of efficiency in DC activation can determine whether sensitization or tolerance is established [128,129].

Furthermore, a dose–response relationship between nickel intake and dermatitis flare-up has been demonstrated [130].

Hongrui Guo et al. demonstrated that NiCl_2_ activates nuclear factor kappa B (NF-κB), mitogen-activated protein kinases (MAPKs), and interferon regulatory factor 3 (IRF3) signaling pathways in primary bone marrow-derived macrophages (BMDMs). This activation leads to altered transcription levels of interleukin-1β (IL1β), -6, -8, and -18; tumor necrosis factor-α (TNF-α); interferon β (INF-β); as well as activated Nod-like receptor 3 (NLRP3) inflammasome pathway [131].

While it has been shown that nickel can induce an inflammatory response, its specific role and the potential association between nickel exposure and the development or exacerbation of IBD remain unclear.

As early as 1988, the potential role of dietary nickel was hypothesized as a possible pathogenic factor in IBD. A German study involving 23 patients with CD who underwent an extended standard patch test revealed a higher incidence of nickel sulfate hypersensitivity compared to healthy controls [132].

A study involving 65 Japanese UC patients and 22 healthy controls who used metallic dental implants and/or prostheses containing nickel demonstrated that UC patients have a significantly higher incidence of hypersensitivity to nickel compared to healthy controls (*p* = 0.0362). Moreover, a higher degree of lymphocyte responsiveness was observed in UC patients in comparison to healthy controls when exposed to metal allergens [133]. This suggests the potential involvement of nickel hypersensitivity in the pathogenesis of UC.

Ogasawara et al. discovered that the concentrations of transition metals, including Fe, Cr, and Co, in the hair of CD patients were significantly higher than those in control subjects. The Ni concentration was slightly higher, and notably, high concentrations of Fe, Cr, Co, and Ni were detected in patients with low serum ferritin levels [134].

Using a novel method of Synchrotron radiation-induced X-ray fluorescence spectroscopy and X-ray absorption fine-structure analysis, a surprisingly high concentration of nickel particles was identified in the submucosa of tissues affected by CD. No other metallic elements were detected alongside nickel [135].

The process of uptake and release of nickel particles within THP-1 cells was observed, highlighting the role of macrophage autophagy mechanisms in particle elimination. This may explain why nickel particles, efficiently eliminated by healthy individuals, cannot be cleared in individuals with specific genetic backgrounds, such as CD patients.

Even more astonishing was the finding that nickel particles induced colitis in mice carrying mutations of the IBD susceptibility protein A20/TNFAIP3. Additionally, nickel particles exacerbated dextran sulfate sodium-induced colitis in mice with myeloid cell-specific Atg5 deficiency [135].

Taken altogether, these findings suggest that the ingestion of nickel particles may exacerbate CD by disrupting autophagic processes in the intestine.

Like the potential role of metal allergy in IBS [136], the pathogenic mechanisms of IBD have now also been suggested to include metal-induced hypersensitivity responses in certain subsets of disease [137]. However, further studies are required to confirm these associations and explore the potential role of a low-nickel diet.

Table 2 summarizes the potential mechanisms underlying the discussed adverse food reactions in patients with IBD.

## 6. Conclusions and Future Directions

In this comprehensive narrative review, we have explored the complex relationship between food intolerances and food allergies in the context of IBD (Figure 2). Several types of food intolerances (such as lactose, fructose, and histamine), as well as nickel allergy and non-celiac gluten sensitivity, can be more prevalent in certain subsets of IBD and can also exacerbate symptoms, particularly during disease flares. This suggests the need for increased clinical awareness and tailored dietary management strategies for this population.

Furthermore, our analysis suggests that food allergies, while not definitively established as more frequent in IBD, might occur more commonly due to increased epithelial barrier permeability, a common feature in IBD pathophysiology. While this connection requires further investigation, it underlines the potential effect of disrupted gut barrier function on the development and exacerbation of food allergies in individuals with IBD.

Future research should prioritize the identification of specific dietary strategies that can effectively mitigate the exacerbation of IBD symptoms in the presence of food intolerances and allergies, as well as the evaluation of the role of the gut microbiota in mediating the complex interaction among food intolerances and allergies and IBD. Additionally, investigating the potential of targeted dietary interventions may offer promising strategies for managing both food intolerances and food allergies in the context of IBD. Moreover, further elucidating the complex mechanisms underlying epithelial barrier integrity and permeability in the setting of IBD could pave the way for innovative approaches aimed at strengthening gut barrier function and reducing the risk of food allergen sensitization.

Collaborative efforts among gastroenterologists, allergists, immunologists, and dietitians are imperative to develop comprehensive, evidence-based guidelines for the management of dietary concerns in individuals with IBD, aiming toward more tailored and effective management strategies.

## 7. Limitations

The comprehension of food intolerances and allergies in the setting of IBD is jeopardized by several limitations. In terms of evidence production, many existing studies have small sample sizes and lack control groups, which hinders the ability to draw definitive conclusions. Additionally, the most commons methods used to evaluate food allergies both in clinical practice and the research setting, such as serum IgE levels and skin prick tests, may not always provide optimal measures for reliable diagnosis. Furthermore, the differentiation between immune-mediated and non-immune-mediated adverse reactions to food can be challenging, impacting the differential diagnosis between functional and organic disorders and the consequent clinical management.

## Figures and Tables

**Figure 1 nutrients-16-00351-f001:**
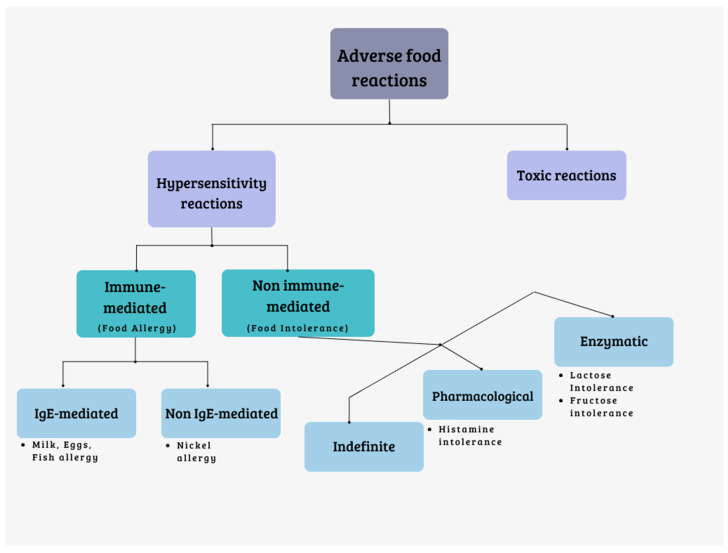
Classification of adverse food reactions.

**Figure 2 nutrients-16-00351-f002:**
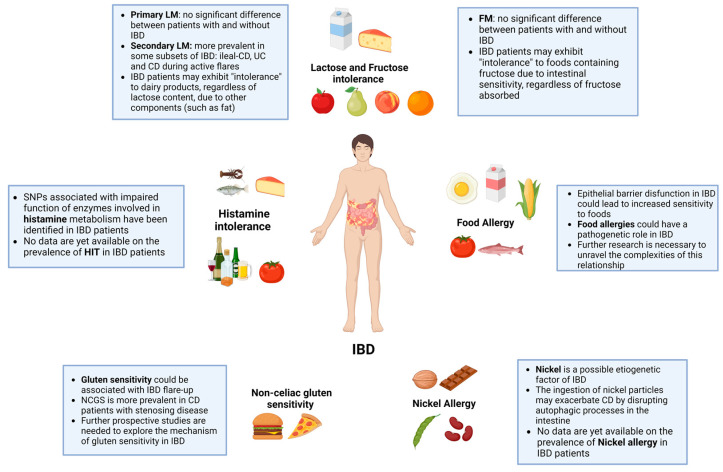
Food intolerances and allergies in IBD.

**Table 1 nutrients-16-00351-t001:** Food allergies and intolerances.

Condition		Mechanism	Common Causative Foods or Substances	References
Food allergy		Reactions mediated by the immune system	Milk, eggs, peanuts, fish, shellfish, plant-based food	[9]
Food intolerance	Enzyme deficiency	Inability to metabolize determined substances present in food	Lactose, fructose, trehalose, sucrose, sorbitol	[13,14]
	Pharmacological intolerance	Direct: caused by chemical substances naturally present in food Indirect: caused by the release of mediators from cells induced by certain foods	Foods rich in histamine (fermented cheeses, salami, oily fish, spinach, tomatoes), methylxanthines (coffee, chocolate, tea, cola), phenylethylamine (cheese, red wine), tyramine (potatoes, cabbage, tuna)	[13,14]
	Indefinite intolerance	Exact mechanism unknown	Food additives (preservatives, dyes, sweeteners, antioxidants), processed foods (canned meat products, syrups, fruit juices)	[13,14]

**Table 2 nutrients-16-00351-t002:** Mechanisms of adverse food reactions in IBD.

Condition	Mechanism	Available Data	References
Lactose intolerance	Damage to the small bowel mucosa with consequent lactase deficiency and lactose malabsorption	Humans	[17,18,28,29]
Fructose intolerance	Inflammation may lead to downregulation of the primary intestinal fructose transporter (GLUT5 receptor)	Rabbits	[57,58]
Histamine intolerance	SNP associated with altered function of enzymes involved in histamine metabolism and reduced DAO activity may lead to HIT	Humans	[97,98]
Non-celiac gluten sensitivity	Fermentable carbohydrates in gluten-containing foods may result in increased intraluminal gas leading to increased gastrointestinal symptoms in IBD patients with stenosing disease	Humans	[99,101]
Nickel allergy	Altered macrophage autophagy mechanisms may lead to accumulation of nickel particles able to worsen colitis	THP-1 cells, murine models	[135]

## Data Availability

Data available in a publicly accessible repository. The data presented in this study are openly available in PubMed.

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
