# Peer review of "Adverse Food Reactions in Inflammatory Bowel Disease: State of the Art and Future Perspectives"

_nutrients, 2024, doi:10.3390/nu16030351_

Round 1

Reviewer 1 Report

Comments and Suggestions for Authors

The review provides an overview of the relationship between food allergies, intolerances, and inflammatory bowel disease (IBD).

Please ensure that the heading format aligns with the style guide or requirements of the journal.

Please ensure consistency in the use of terminology and formatting throughout the text.

The introduction effectively addresses the challenge of linking symptoms to food intake, especially in patients with inflammatory bowel disease (IBD).

Please consider streamlining sentences (in the introduction and in the text) for more clarity and readability.

The introduction could conclude with a sentence that hints at the potential implications of the study's findings and suggests future research directions.

Chapter 2. The information is logically organized, starting with the classification of adverse reactions and then delving into the specifics of food allergies and intolerances. While cross-reactivity is mentioned briefly, providing a bit more depth on this aspect could enrich the discussion, including examples or common scenarios. The prevalence data for food allergies is mentioned but may benefit from including more recent statistics to reflect the current landscape.

Chapter 3. Maintain consistency in the use of terminology. For example, "lactose intolerance" and "lactose malabsorption" are used interchangeably in the text. Clarify and use the terms consistently to avoid confusion.

Chapter 4. The section discussing the murine model exposure to ovalbumin could be expanded to provide a more detailed explanation of the experimental results. Clarify how these results contribute to the understanding of the relationship between food allergies and IBD. Also, the section discussing the variability in sensitivity to foods among patients with UC and CD introduces interesting findings. However, it would be beneficial to explore potential reasons for these variations, such as genetic factors, environmental influences, or the heterogeneity of IBD itself.

The conclusion provides a good recapitulation of the key findings. However, consider condensing the recap for a more concise and impactful summary. Emphasize the main takeaways without repeating detailed information.

Comments on the Quality of English Language

Minor editing of English language required

Author Response

Response to Reviewer 1

Thank you for dedicating time to assess our manuscript. Our group kindly thanks the editors and reviewers for their consideration and valuable comments. The feedback provided by the reviewers has been instrumental in enhancing the quality of our manuscript, and we have made revisions accordingly.

Below, you will find a detailed response addressing the raised issues.

The review provides an overview of the relationship between food allergies, intolerances, and inflammatory bowel disease (IBD).

  1. Please ensure that the heading format aligns with the style guide or requirements of the journal.

Response: Thank you for your valuable feedback. We have carefully revised the heading format to align it with the style guide and journal requirements. We have meticulously recompiled the literature in adherence to the journal's guidelines, ensuring proper formatting in both the main text and the references table (we have included dois for all references).

  1. Please ensure consistency in the use of terminology and formatting throughout the text.

Response: Thank you for your suggestion. We have ensured consistency in the use of terminology and formatting throughout the entire manuscript.

The introduction effectively addresses the challenge of linking symptoms to food intake, especially in patients with inflammatory bowel disease (IBD).

  1. Please consider streamlining sentences (in the introduction and in the text) for more clarity and readability.

Response: Thank you for your thoughtful evaluation. We have made significant efforts to streamline sentences throughout the introduction to enhance clarity and readability (lines 45-53).

  1. The introduction could conclude with a sentence that hints at the potential implications of the study's findings and suggests future research directions.

Response: Thank you for your suggestion. We have carefully revised the introduction to incorporate a concluding sentence that hints at the potential implications of our study's findings and suggests future research directions (lines 66-70).

Chapter 2. The information is logically organized, starting with the classification of adverse reactions and then delving into the specifics of food allergies and intolerances.

  1. While cross-reactivity is mentioned briefly, providing a bit more depth on this aspect could enrich the discussion, including examples or common scenarios. The prevalence data for food allergies is mentioned but may benefit from including more recent statistics to reflect the current landscape.

Response: Thank you for your detailed review of Chapter 2. We appreciate your positive feedback on the logical organization of information. Based on your suggestion, we have expanded the discussion on cross-reactivity, providing more depth, including examples and common scenarios to enrich the content (lines 91-96). Additionally, we have updated the prevalence data for food allergies to include more recent statistics, ensuring a reflection of the current landscape (lines 85-86).

  1. Chapter 3. Maintain consistency in the use of terminology. For example, "lactose intolerance" and "lactose malabsorption" are used interchangeably in the text. Clarify and use the terms consistently to avoid confusion.

Response: Thank you for review of Chapter 3. The distinction between the two conditions are clarified at the beginning of the paragraph (lines 135-139). In accordance with your guidance, we have made every effort to maintain consistency in the use of terminology throughout the lactose and fructose sections (Chapter 3.1 – 3.2).

  1. Chapter 4. The section discussing the murine model exposure to ovalbumin could be expanded to provide a more detailed explanation of the experimental results. Clarify how these results contribute to the understanding of the relationship between food allergies and IBD. Also, the section discussing the variability in sensitivity to foods among patients with UC and CD introduces interesting findings. However, it would be beneficial to explore potential reasons for these variations, such as genetic factors, environmental influences, or the heterogeneity of IBD itself.

Response: Thank you for your insightful comments on Chapter 4. We appreciate your feedback regarding the murine model exposure to ovalbumin, and we have expanded this section to provide a more detailed explanation of the experimental results (lines 462-470). Moreover, your observation about the variability in sensitivity to foods among patients with UC and CD has been duly noted. In response, we have enriched the discussion by exploring potential reasons for these variations (lines 500-506).

The conclusion provides a good recapitulation of the key findings.

  1. However, consider condensing the recap for a more concise and impactful summary. Emphasize the main takeaways without repeating detailed information.

Response: Thank you for your feedback. We have revised the conclusion, emphasizing the main takeaways without redundant details (Chapter 5). We believe these changes contribute to a more streamlined and impactful summary of the study's key findings.

Reviewer 2 Report

Comments and Suggestions for Authors the review is well written and the subject is very interesting albeit less frequently studied. nonetheless, these are my remarks:
  1. the article is longer than expected and it has too many general terms defined and described. it can be difficult to read and it resembles more to a book chapter rather than an article. 
  2. the methodology used in documenting this review is not described. 
Comments on the Quality of English Language

Minor revision

Author Response

Thank you for dedicating time to assess our manuscript. Our group kindly thanks the editors and reviewers for their consideration and valuable comments. The feedback provided by the reviewers has been instrumental in enhancing the quality of our manuscript, and we have made revisions accordingly.

Below, you will find a detailed response addressing the raised issues.

The review is well written and the subject is very interesting albeit less frequently studied. nonetheless, these are my remarks:

  1. The article is longer than expected and it has too many general terms defined and described. it can be difficult to read and it resembles more to a book chapter rather than an article.

Response: Thank you for your positive feedback on the review and your valuable remarks. We have revised both the introduction and the main text, streamlining sentences for more clarity and readability. Additionally, to enhance the overall appeal and accessibility of the manuscript, we have incorporated two concise tables (Table 1, Line 127 – Table 2, Line 578) that provide a summary of key information.

  1. The methodology used in documenting this review is not described.

Response: Thank you for your suggestion. We have now included a section outlining the methodology used in this review (Chapter 7, Lines 619-630).

Reviewer 3 Report

Comments and Suggestions for Authors

I read with interest the article " Lactose, food intolerances and allergies in Inflammatory Bowel Disease: state of the art and future perspectives." This is a very important topic for both clinicians and patients themselves.

The manuscript is correctly structured, but here are some of my comments:

- The key words should be arranged alphabetically.

- The English language should be improved.

- The literature should be compiled according to the guidelines of the journal, both in the text and in the table of references (it is worth adding doi as well).

- In the introduction, it is worth starting with a description of UC and CD, followed by their symptoms.

- In Chapter 2, it is worth adding a table summarising food allergies and intolerances (3 categories) and clearly indicating the differences between them to make the manuscript more attractive to potential readers.

- It would also be worth considering a summary table after Chapters 3 and 4 with an indication of the mechanisms of food intolerances of individual dietary components and IBD, e.g. lactose intolerance and IBD.

- It would also be useful to add limitations.

Comments on the Quality of English Language

The English language should be improved.

Author Response

Response to Reviewer 3

Thank you for dedicating time to assess our manuscript. Our group kindly thanks the editors and reviewers for their consideration and valuable comments. The feedback provided by the reviewers has been instrumental in enhancing the quality of our manuscript, and we have made revisions accordingly.

Below, you will find a detailed response addressing the raised issues.

I read with interest the article " Lactose, food intolerances and allergies in Inflammatory Bowel Disease: state of the art and future perspectives." This is a very important topic for both clinicians and patients themselves. The manuscript is correctly structured, but here are some of my comments:

  1. The key words should be arranged alphabetically.

Response: Thank you for pointing this out. We have arranged the key terms alphabetically (Line 27).

  1. The English language should be improved.

Response: Thank you for your feedback. We have thoroughly reviewed and revised the manuscript to enhance the clarity and coherence of the English language.

  1. The literature should be compiled according to the guidelines of the journal, both in the text and in the table of references (it is worth adding doi as well).

Response: Thank you for your valuable feedback. We have meticulously recompiled the literature in adherence to the journal's guidelines, ensuring proper formatting in both the main text and the references table (we have included dois for all references).

  1. In the introduction, it is worth starting with a description of UC and CD, followed by their symptoms.

Response: Thank you for your suggestion. We have carefully revised the introduction to begin with a brief description of UC and CD followed by their main symptoms (lines 30-34).

  1. In Chapter 2, it is worth adding a table summarising food allergies and intolerances (3 categories) and clearly indicating the differences between them to make the manuscript more attractive to potential readers.

Response: Thank you for your insightful comments on Chapter 2. We really appreciate your input, and we have included a table that summarizes food allergies and intolerances. This table provides a clear overview of the distinctions between the 3 categories, with the aim of enhancing the manuscript's appeal to readers (Table 1, Line 127).

  1. It would also be worth considering a summary table after Chapters 3 and 4 with an indication of the mechanisms of food intolerances of individual dietary components and IBD, e.g. lactose intolerance and IBD.

Response: Thank you for your suggestion. We have created a table that outlines the main mechanisms of the discussed food intolerances and allergies associated with individual dietary components and IBD (Table 2, Line 578).

  1. It would also be useful to add limitations.

Response: Agree. We have incorporated a dedicated section that discusses the limitations of our study (Chapter 6, Lines 609-618).

Round 2

Reviewer 1 Report

Comments and Suggestions for Authors

I suggest moving the methodology after the introduction.

The paper was improved, and the author addressed all reviewer's comments accordingly. 

Author Response

Thank you for your suggestion. We have moved the methodology section after the introduction as recommended. 

Thank you for your careful review.

Reviewer 3 Report

Comments and Suggestions for Authors

The manuscript has been revised as suggested. 

It needs only minor corrections of English.

Comments on the Quality of English Language

-

Author Response

Thank you for your feedback. We have thoroughly reviewed the manuscript and implemented the necessary corrections.

Thank you for your careful review.